

# Successful promotion of physical activity among students of medicine through motivational interview and Web-based intervention

Dubravka Mandic[1], Vesna Bjegovic-Mikanovic[1], Dejana Vukovic[1], Bosiljka Djikanovic[2], Zeljka Stamenkovic[2] and Nebojsa M. Lalic[3]

[1] University of Belgrade, Faculty of Medicine, Centre School of Public Health and Management, Belgrade, Serbia
[2] University of Belgrade, Faculty of Medicine, Institute of Social Medicine, Belgrade, Serbia
[3] University of Belgrade, Faculty of Medicine; Clinic for Endocrinology, Diabetes and Metabolic Diseases, Belgrade, Serbia

Corresponding author
Vesna Bjegovic-Mikanovic,
vesna.bjegovic-mikanovic@med.bg.ac.rs

## ABSTRACT

**Background:** Regular physical activity supports healthy behavior and contributes to the reduction of preventable diseases. Students in their social transition period are the ideal groups for interventions. The higher education period, associated with demanding changes and poor time management, results in a low level of physical activity. In this age, social media usually are a suitable channel of communication and multicomponent interventions are the most desirable. It has not been sufficiently investigated how effective a Web-based approach is among university students when it comes to physical activity in the long-term period. We combined a Web-based approach with motivational interviews and tested these two interventions together and separate to assess their impact on improving the physical activity of medical students 1 year after the intervention.

**Methods:** All 514 first-year students at the Faculty of Medicine in Belgrade were invited to fill in a baseline questionnaire. Also, they underwent measurement of weight, height and waist circumference. After that, students selected a 6 months intervention according to their preference: Intervention through social media (Facebook) (Group 1) or combined with a motivational interview (Group 2). Group 3 consisted of students without any intervention. One year after completion of the 6 months intervention period, all students were invited to a second comprehensive assessment. Analyses were performed employing a wide range of statistical testing, including direct logistic regression, to identify determinants of increased physical activity measured by an average change of Metabolic Equivalent of Task (MET). This outcome measure was defined as the difference between the values at baseline and one year after completion of the 6 months intervention period.

**Results:** Due to a large number of potential determinants of the change of MET, three logistic regression models considered three groups of independent variables: basic socio-demographic and anthropometric data, intervention and willingness for change, and health status with life choices. The only significant model comprised parameters related to the interventions ($p < 0.001$). It accurately classified 73.5% of cases. There is a highly significant overall effect for type of intervention (Wald = 19.5, df = 2, $p < 0.001$) with high odds for the increase of physical activity. Significant

relationship between time and type of intervention also existed ($F = 7.33$, $p < 0.001$, partial $\eta^2 = 0.091$). The influence of both factors (time and interventions) led to a change (increase) in the dependent variable MET.

**Conclusion:** Our study confirmed the presence of low-level physical activity among students of medicine and showed that multicomponent interventions have significant potential for positive change. The desirable effects of the Web-based intervention are higher if an additional booster is involved, such as a motivational interview.

## INTRODUCTION

Regular physical activity supports healthy behavior in general and, together with other healthy lifestyles, contributes to the reduction of preventable diseases and premature mortality (*Blair & Morris, 2009*). On the other hand, physical inactivity is one of the leading risk factors for non-communicable diseases. According to the most recent estimates, the prevalence of insufficient physical activity is 27.5% at the global level (*Guthold et al., 2018*). In 2017, the Institute for Health Metrics and Evaluation (IHME) reported that low physical activity as a global risk factor contributes to ischemic heart disease (in average by 9.1%), stroke (by 4.1%), and diabetes (by 2.8%) (*Institute for Health Metrics & Evaluation (IHME), 2017*).

There are many arguments to recommend improved physical activity: prevention of cognitive function decline, improvement of musculoskeletal function, and maintaining healthy body weight (*Warburton, Nicol & Bredin, 2006*; *Reiner et al., 2013*; *World Health Organization, 2018*) but also feeling better in everyday life (*Lavie, Ozemek & Kachur, 2019*). The World Health Organization recommends daily moderate physical activity of at least 30 min (*World Health Organization, 2010*), and the American Heart Association has even more precise recommendations (American Heart Association Recommendations for Physical Activity in Adults, 2014, https://www.heart.org/en/healthy-living/fitness/fitness-basics/aha-recs-for-physical-activity-in-adults).

Young people between 18 and 35 years are one particular segment of the population that is at risk due to lower levels of physical activity and a sedentary lifestyle. This age-group is characterized as a high transition period, when serious changes occur in life: going for higher education, starting a career, and establishing a family. Some authors consider these demanding elements of life as contributing factors in reducing physical activity (*Deliens et al., 2015*; *Aceijas et al., 2017*). Notably, the higher education period, associated with growing independence, often results in a change of diet, but also a low level of physical activity (*Plotnikoff et al., 2015*). The steepest decline in physical activity is seen at the beginning of the university or college period (*Kwan et al., 2012*; *Sigmundova et al., 2013*). Some studies suggest that participation in public health interventions is useful and can

result in enhanced physical activity and a healthier lifestyle at a later age (*Brynteson & Adams, 1993*; *Friedman et al., 2008*).

According to Plotnikoff,students are the ideal group for interventions fostering an improved lifestyle (*Plotnikoff et al., 2015*). This population is exposed to a dedicated learning environment and represents a numerically significant proportion of the mature population, surrounded by educators who can help to promote a healthy lifestyle.

The well-known general strategies for behavior change are (1) group interventions, (2) individual interventions, (3) computer-technological interventions, and (4) multicomponent interventions (*Rollnick, Miller & Butler, 2007*). The most successful are multicomponent or blended interventions, which combine different strategies (*Elvsaas et al., 2017*). Also, programs with pre-arranged frequent contacts are far more successful than interventions offering one contact only (*Greaves et al., 2011*).

Recently, motivational interviews gained popularity in different settings (*Karnes et al., 2015*). This technique initially developed to overcome the behavior of addiction (drugs, alcohol), has spread on many fields that require motivation to change behavior (*Hettema, Steele & Miller, 2005*). A 2010 meta-analysis shows that a motivational interview can lead to behavior change in different groups: firefighters, smokers, students, as well as members of different ethnic groups (*Lundahl et al., 2010*). Also, this approach has been proven to be effective in the improvement of physical activity (*Rubak et al., 2005*). Motivational interviews raise self-dedication, provide competences, decrease ambivalence, stimulate intrinsic motivation, and increase the possibility of positive change in physical activity. Systematic reviews pointed on the strengths of motivational interviews for empowering and encouraging young people to enhance health behaviors (*Mutschler et al., 2018*).

Besides the motivational interview, the use of the Internet and Web-based technologies as part of interventions has several advantages: access to a large number of respondents at minimal cost, availability of intervention at any place any time, and the possibility of receiving feedback as well as providing personalized information (*Lustria et al., 2013*), for a group of young people, the most accessible form of communication (*Laranjo et al., 2015*; *Afshin et al., 2016*). One meta-analysis confirmed that internet interventions could lead to significant improvement in health outcomes (*Lustria et al., 2013*). Some studies suggest that the use of the internet, and primarily Web-based social networks such as Facebook, helps to improve physical activity (*Maher et al., 2015*; *Zhang et al., 2016*). Web-based social networks are well accepted in the young generation socialized in the modern environment of the Web and multilateral exchange of experience and worries (*Suner, Yilmaz & Pişkin, 2019*). Unlike traditional interventions, online social networks typically achieve high levels of user engagement in physical activity and retention (*Davies et al., 2012*). The internet use and other accompanying indications request adequate eHealth literacy (*Mitsuhashi, 2018*) and particularly among young adults. As an example, in the web-based intervention, it may be worthwhile to consider whether the motivational images are expected to increase physical activity or not (*Prichard et al., 2020*). There is research that shows that sometimes motivational images on social media have the opposite of the intended effect.

However, despite its potential for behavior change and immense popularity, it has not been sufficiently investigated how effective a Web-based approach is among university students when it comes to physical activity (*Maher et al., 2014*). Notably, students of medicine, due to the time-consuming curricula, are one of the most sedentary categories among university students (*Lee & Graham, 2001*). Additionally, students of medicine are a crucial strategic group because intervening in their early twenties likely will exert a lasting impact on their future careers as medical practitioners, clinicians, or academic teachers and public health promoters.

The objective of this study was to determine the effects of different interventional strategies on improving the physical activity of medical students and to assess predictors of possible improvement. We hypothesized that students exposed to a combined intervention of motivational interview and Web-based intervention during 6 months would show stronger long-term effects on the physical activity one year later than those exposed to Web-based intervention only or no intervention at all. Also, we expected positive changes in physical parameters, such as body mass index and waist circumference.

## MATERIALS AND METHODS

### Ethical approval

This study has obtained ethical approval by the Ethical Board of the University of Belgrade, Faculty of Medicine (decision No: 29/IX-7, date: September 19, 2016).

### Study design

The approach to research was a prospective cohort study among first-year medical students at the Faculty of Medicine, University of Belgrade, before and one year after a six months intervention to improve their physical activity. Also, the approach embraced action research and participation in activities designed to have immediate effects on behavior, changes in lifestyles, and particularly physical activity in the community of first-year students.

### Population, recruitment of participants and procedures

The targeted population presented all first-year students of the Faculty of Medicine, in total 514. We started recruiting students during regular teaching hours at the beginning of the academic year 2016/2017. All students received information (through a flyer or blackboard announcement) about the research topic and were invited to attend a class presenting the research in detail. We obtained the written informed consent from the participants prior to data collection. A total of 254 (49.4%) students filled in correctly both the informed consent template and a baseline questionnaire (see Annex 1), provided contact details and underwent measurement of weight, height and waist circumference. Completing the questionnaire was voluntary, anonymous and without incentives.

The students were then invited to enter a self-selected 6 months intervention according to their preference, either Web-based intervention through Facebook only (Group 1) or

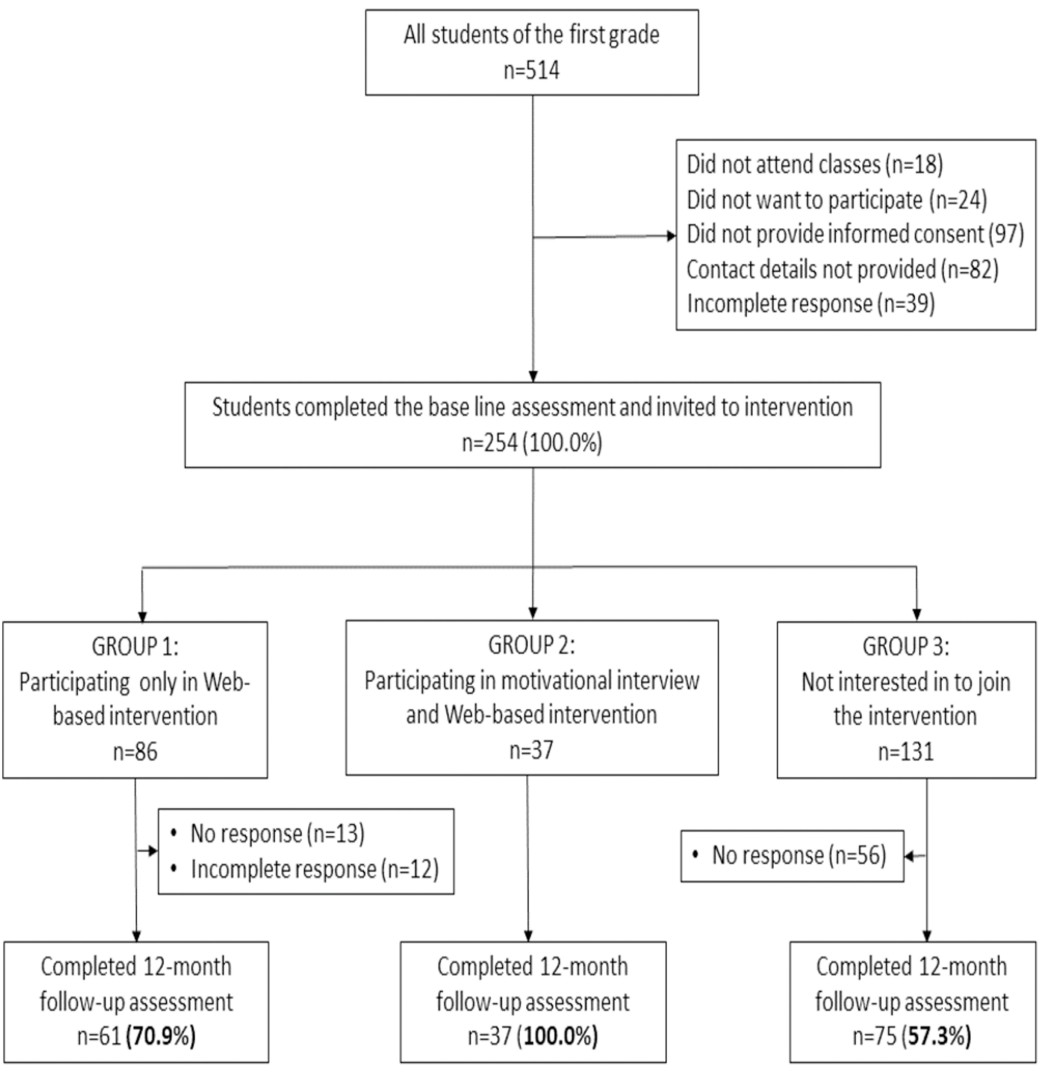

**Figure 1 Participants' flowchart—recruitment, choice of intervention group and retention.**

combined intervention with a motivational interview and Web-based intervention through Facebook (Group 2): 86 students chose Group 1, and 37 students selected Group 2. Students who were not interested in joining the intervention formed Group 3, consisting of 131 students. Figure 1 presents the flowchart of participants' recruitment and retention. We did not form a group with the motivational interview alone: (a) because of the limited number of participants, and (b) because our research question was to see whether the common Web-based intervention can be improved by motivational interviews.

After 1 year from completion of the six months intervention, all students were invited to a second assessment. In Group 1, the response rate was 70.9%, and in Group 2 100%, while among those who were not interested in joining any intervention (Group 3), the response rate was 57.3%. We choose this extended period to be sure about a long-term impact on lifestyle.

## Research instruments

At baseline, the students filled in the questionnaire designed to create a physical activity profile with a provision of personalized feedback. Before the main research, a pre-test and a series of focus groups, including 15 medical students, each served to determine the feasibility of the research in terms of the amount and quality of data to be collected. Students gave feedback about the content of the questionnaire, the relevance of the questions, and its length. Based on their opinions and recommendations, the final version of the questionnaire was optimized.

The questionnaire contained the following sections: (1) socio-demographics, anthropometric data, and self-rated health; (2) the current level of physical activity; (3) the preferred type of physical activity; (4) multidimensional well-being; (5) mental health; and (6) lifestyle. The completion of the questionnaire (Annex 1) took approximately 30 min. An introductory page presented the study. It contained information about the research goals and the secret identification code that was used to match data from the first and second waves of this study.

Within the socio-demographics, we included: gender, age, nationality, type of transportation to the Faculty, time to get to the Faculty, academic achievements (average mark), height, weight, waist circumference, relationship status, and household income per capita. The majority of the items in this section were drawn from the European Health Interview Survey wave 2 (EHIS wave 2) (*Eurostat, 2013*). The anthropometric data were assessed in a separate room with the help of at least one medical doctor to be as precise as possible.

Physical activity (PA) was examined at baseline through the International Physical Activity Questionnaire – Short Version (IPAQ-SV) (https://sites.google.com/site/theipaq/) (*Craig et al., 2003*). The data was then used to estimate total weekly physical activity by weighting the reported minutes per week within each activity category by a Metabolic Equivalent of Task (MET) energy expenditure estimate assigned to each category of activity. The weighted MET-minutes per week were calculated as duration of activity multiplied by frequency per week multiplied by 8 for vigorous-intensity PA, by 4 for moderate-intensity, and by 3.3 for walking level. The total number of MET-minutes per week was obtained by summing up the subtotals. This value was selected as the dependent variable. The section about physical activity also presented sub-sections with consideration of the actual level of physical activity, satisfaction with the level of physical activity, reasons for inactivity, wish to improve physical activity. In case the respondent expressed a wish to be more active, the following page included a modified version of the IPAQ-SV. The result of this IPAQ-SV was the total number of wished MET-minutes per week.

The next section included two items regarding the preferred type of physical activity and the type of physical activity to be improved in the next month. Both items present the same options, with the possibility to select more answers: walking/hiking, running/jogging, individual gym workout, group gym workout, indoor/outdoor group sports, water sports, and winter sports, dance, other-specify.

The next section included items regarding mental health, and it contains an 8-items measure of depression symptoms from the Physical Health Questionnaire—8 (PHQ-8) (*Kroenke et al., 2009*; *McMahon et al., 2017*).

The last set of questions was about self-rated health and life choices, and it included: smoking cigarets, cannabis/marijuana consumption, binge drinking, and intakes of fruit and vegetable.

## Intervention for improvement of physical activity

After completing the baseline study, the participants were invited to join a specially created Web-based project group through Facebook. The project group aimed to continuously promote healthy lifestyles and motivate students to engage in physical activity by sharing various appropriate content based on similar interventions (*Bonnie, Stroud & Breiner, 2014*).

The Web-based (WB) intervention in both groups (Group 1 and 2) was "closed," allowing only participants enrolled in the study to view the posted information and the profiles of other participants who are members of the group. The Web-based intervention in each group (1 and 2) had three administrators (public health experts). Their tasks were to post and possibly moderate the comment section for each intervention content, referring to certain common physical activities—events such as participation in the Marathon in Belgrade; regularly check the main Facebook page of the group, to decide whether the content that the students have set up desirably represents the group; and remove or prohibit group members showing offensive behavior. The Web-based intervention had three components designed to increase physical activity, and included:

1. Scientific publications regarding physical activity promotion posted on the group Facebook wall (once a week, in total 24 articles selected by administrators); these works highlighted the low physical activity levels of the student population, with the associated disease burden, and addressed prevalent socio-cultural norms and barriers to physical activity commonly reported among students.

2. Questions leading to a discussion on topics related to physical activity and participant engagement (once a week, in total 24 questions opening discussions in which each student from Group 1 and 2 has an opportunity to participate and express opinions); the purpose of the weekly discussion topics was sharing of personal experiences with physical activity and giving/receiving social support for physical activity (e.g., "What are your thoughts on the physical activity statistics among students? How can you incorporate more physical activity into your daily routine?").

3. Motivational text messages and images promoting physical activity (three times a week, in total 72 posts); these text messages acted as another mechanism of social support and provided: (a) tips on strategies to increase physical activity throughout the day (e.g., "Set aside time today for several 10–15-min walks. Walking 30 min at a moderate-intensity on 5 days each week = 150 min!"), (b) information on how to overcome barriers to physical activity (e.g., "Don't let the lack of time interfere with your physical activity routine. Take a walk with your whole family this weekend".),
(c) reminders on the health benefits of physical activity (e.g., "Physical activity promotes health and reduces the risk of bone fractures and osteoporosis."), and (d) motivational/inspirational tips and quotes to participants (e.g., " 'Each person must live their life as a model to others'.—Rosa Parks").

The aim of this closed discussion via Facebook was to motivate mutually by the exchange of positive experiences and intentions to improve physical activity.

Students additionally got an opportunity to apply for a motivational interview, and 37 students accepted. Therefore they formed a Group 2 exposed both to Web-based intervention and motivational interview. Motivational interviews were done using Motivational Interviewing techniques (*Miller & Rollnick, 2013*). Individual consulting within a motivational interview was a voluntary activity, and its content is treated as confidential and anonymous. The interviewers were health workers who were trained by a certified trainer for motivational interviewing Motivational Interviewing Network of Trainers (*MINT, 2016*). Motivational interviews were conducted at a time agreed with a certified teacher, in an area that can provide undisturbed individual work. Each student attended two face-to-face sessions at the beginning of the intervention, individually tailored, with a duration of 45 min each. The content and the spirit of the sessions are presented in the Protocol for Motivational Interview (see Annex 2). The second session followed after 14 days and served as self-reflection on the feedback for the physical activity, geared towards strengthening commitment to make a positive change.

### Statistical analysis

The variables in this study are presented as continuous data (scale) or categorical (nominal and ordinal). Continuous data are summarized as mean value with standard deviation and 95% CI or for skewed data as median with inter quartile range (IQR). Categorical data are provided by their absolute numbers and percentages. In the analysis Chi-square test, Student's *t*-test for independent or paired samples, Mann–Whitney test (not Gaussian distribution), Kruskal–Wallis test, Fisher's Exact test (when necessary, in the case of low frequencies), Wilcoxon signed ranks test, ANOVA and post hoc analysis, and binary logistic regression (Enter model) were used. The alpha level at 0.05 (*p* value < 0.05) was considered to indicate significance. Levene's test was employed to assume the Equality of Variances. The normal distribution of each variable was to be checked with the Kolmogorov–Smirnoff test. To perform mixed-design analysis of variance (ANOVA) for repeated measurements, a logarithmic transformation of data for MET was performed to achieve normal distribution of samples in all (three) examined groups for both measurement times (baseline and after 12 months). Atypical outliers were eliminated, and the selected significance level was set to $\alpha = 0.01$.

## RESULTS

The descriptive variables of the study participants are presented in Table 1. At the end of the intervention period, in total, 173 medical students remained in the three groups. The first group of "Web-based intervention only" had 61 students, the second group with

**Table 1 Characteristics of study participants at baseline.**

| Characteristics | Total | Group 1 | Group 2 | Group 3 | p |
|---|---|---|---|---|---|
| Number | 173 (100.0%) | 61 (35.3%) | 37 (21.4%) | 75 (43.4%) | – |
| Gender[A] | | | | | |
| Male, n (%) / | 51 (29.5%) / | 14 (23.0%) / | 14 (37.8%) / | 23 (30.7%) / | 0.280 |
| Female, n (%) | 122 (70.5%) | 47 (77.0%) | 23 (62.2%) | 52 (69.3%) | |
| Age, years (SD)[B] | 20.34 ± 0.57 | 20.28±0.52 | 20.46 ± 0.69 | 20.33 ± 0.55 | 0.319 |
| Academic achievement, average mark (SD)[B] | 8.70 ± 0.80 | 8.80 ± 0.79 | 8.60 ± 0.71 | 8.66 ± 0.84 | 0.402 |
| Type of transportation to the faculty[C] | | | | | |
| Walk | 30 (17.3%) | 10 (16.4%) | 9 (24.3%) | 11 (14.7%) | 0.725 |
| Bicycle | 0 (0.0%) | 0 (0.0%) | 0 (0.0%) | 0 (0.0%) | |
| Car | 2 (1.2%) | 1 (1.6%) | 0 (0.0%) | 1 (1.3%) | |
| Public transport | 141 (81.5%) | 50 (82.0%) | 28 (75.7%) | 63 (84.0%) | |
| Time to get to the faculty, median (IQR)[D] | 25 (15) | 30 (13) | 20 (15) | 30 (15) | 0.103 |
| Household income per capita[D] | | | | | |
| Less than 300€ | 52 (30.1%) | 18 (29.5%) | 14 (37.8%) | 20 (26.7%) | 0.677 |
| 300–400€ | 55 (31.8%) | 21 (34.4%) | 8 (21.6%) | 26 (34.7%) | |
| 400–500€ | 28 (16.2%) | 10 (16.4%) | 8 (21.6%) | 10 (13.3%) | |
| 500–600€ | 17 (9.8%) | 4 (6.6%) | 5 (13.5%) | 8 (10.7%) | |
| >600€ | 21 (12.1%) | 8 (13.1%) | 2 (5.4%) | 11 (14.7%) | |

Note:
Data are presented as n (%), means ± standard deviation or median (IQR); not significant (NS) between groups for all parameters. A: Tested by Chi-Square test; B: Tested by One-Way ANOVA; C: Tested by Fisher's Exact test; D: Tested by Kruskal-Wallis test.

both "Motivational Interview and Web-based intervention"—37 and the group without intervention comprised 75 students (see Fig. 1). There was no significant difference between the three groups at the baseline assessment.

Students' average age was 20.34 years (SD 0.57), among which 51 (29.5%) were male, while almost two-thirds of them were female 122 (70.5%). Their average academic achievement was 8.704 (SD 0.80) out of 10. Results are showing that the majority of them are using public transport, while only 17.3% are walking to the faculty. Bicycles and cars were only marginal choices. Participants reported that the usual time to get to the faculty was 25 min. On average, household income per capita per month was 300€, while only 12.1% of students reported more than 600€.

Health status and life choices are presented in Table 2. Overall, most of the students estimated their health as very good (81/46.8%) and good (68/39.3%), while 22 (12.7%) rated their health as average. The majority of students had good mental health, with a PHQ-8 score below 10. Students had an average BMI of 21.62 kg/m² with SD 2.75, being in a range of normal weight. Average waist circumference was normal, for both male and female group of students. The results related to smoking were: 15.6% of students declared actual smoking of cigarets, and 5.2% used cannabis/marijuana in the year preceding the study. Half of the students (50.3%) reported binge drinking (6 or more drinks on one occasion) during the year before the study. In terms of healthy eating habits, daily intake of fruit and vegetable was adequate only for one-third of the participants. There was no significant difference in examined parameters between groups.

**Table 2 Health status and life-choices of medical students at baseline.**

| Characteristics | Total | Group 1 | Group 2 | Group 3 | p |
|---|---|---|---|---|---|
| Number | 173 (100.0%) | 61 (35.3%) | 37 (21.4%) | 75 (43.4%) | |
| Self-rated health, n (%)[D] | | | | | |
| Very good | 81 (46.8%) | 32 (52.5%) | 12 (32.4%) | 37 (49.3%) | 0.168 |
| Good | 68 (39.3%) | 25 (41.0%) | 20 (54.1%) | 23 (30.7%) | |
| Average | 22 (12.7%) | 3 (4.9%) | 5 (13.5%) | 14 (18.7%) | |
| Bad | 1 (0.6%) | 1 (1.6%) | 0 (0.0%) | 0 (0.0%) | |
| Very bad | 1 (0.6%) | 0 (0.0%) | 0 (0.0%) | 1 (1.3%) | |
| Mental health, good (PHQ-8 < 10), n (%)[A] | 148 (85.5%) | 55 (90.2%) | 31 (83.8%) | 62 (82.7%) | 0.439 |
| Body mass measures[B] | | | | | |
| BMI (kg/m$^2$) (mean ± SD) | 21.62 ± 2.75 | 21.40 ± 2.37 | 21.86 ± 2.70 | 21.67 ± 3.06 | 0.710 |
| Waist circumference—male (cm) (mean 95% CI)[B] | 80.5 [78.1–82.9] | 80.2 [78.8–88.0] | 79.1 [74.2–86.0] | 79.1 [75.8–82.4] | 0.333 |
| Waist circumference—female (cm) (mean 95% CI)[B] | 72.0 [70.46–73.4] | 73.9 [71.4–76.4] | 71.3 [68.0–74.6] | 70.5 [68.7–72.4] | 0.100 |
| Smoking, n (%)[A] | 27 (15.6%) | 10 (16.4%) | 7 (18.9%) | 10 (13.3%) | 0.729 |
| Marijuana use in last 12 months, n (%)[C] | 9 (5.2%) | 2 (3.3%) | 1 (2.7%) | 6 (8.0%) | 0.447 |
| Binge drinking in last 12 months, n (%)[A] | 87 (50.3%) | 30 (49.2%) | 17 (45.9%) | 40 (53.3%) | 0.746 |
| Consumption of fresh fruits daily, n (%)[A] | 60 (34.7%) | 21 (34.4%) | 12 (32.4%) | 27 (36.0%) | 0.936 |
| Consumption of vegetables daily (%)[A] | 61 (35.3%) | 23 (37.7%) | 14 (37.8%) | 24 (32.0%) | 0.785 |

**Note:**
Data are presented as n (%) or means ± standard deviation or means (95CI intervals); Not significant (NS) between groups for all parameters; A: Tested by Chi-Square test; B: Tested by One-Way ANOVA; C: Tested by Fisher's Exact test; D: Tested by Kruskal-Wallis test.

Comparison of physical activity calculated in average MET (Table 3) 12 months after the intervention period has shown a significant difference between both groups with the intervention compared to Group 3 without any intervention ($p < 0.001$) (Kruskal–Wallis test). Level of physical activity, measured in median MET (IQR) has increased from 1,506 (2,058) to 2,813 (1,680) ($p < 0.001$) in Group 1 exposed only to Web-based intervention, and from 1,386 (1,579) to 2,586 (1,794) ($p < 0.001$) in Group 2 exposed to both Web intervention and motivational interview. At the same time, in Group 3, without intervention, starting from baseline median MET (IQR) of 1,155 (1,053), physical activity has been slightly increased. However, this change is not significant, as shown in Table 3.

Most of the examined participants were not satisfied at the beginning of the study with the level of their physical activity, without significant difference between the groups. The level of satisfaction did not significantly increase 12 months after intervention ($p = 0.287$) for any of the three groups. The main reason leading to physical inactivity was the lack of time for all participants, without significant difference between the groups. Then tiredness and unwillingness followed as reasons. We did not find any significant difference between groups, except for one reason, "no wish for physical activity," which was the most present in Group 2 ($p = 0.012$). In terms of the type of preferred activity, there is a significant difference ($p = 0.023$) between all three groups for walking after the intervention. The highest percentage of students who prefer walking after 12 months was in Group 2. A wish for physical activity is present at the beginning of the study period for

**Table 3 The physical activity and related variables before and after intervention.**

| Characteristics | Group 1 | | Group 2 | | Group 3 | | *p* BL | *p* 12M |
|---|---|---|---|---|---|---|---|---|
| | BL | 12M | BL | 12M | BL | 12M | | |
| Physical activity, median MET (IQR) | 1,506 (2,058) | 2,813 (1,680)* | 1,386 (1,579) | 2,586 (1,794)* | 1,155 (1,053) | 1,222 (1,253) | 0.398[D] | *p* < 0.001[D] |
| Total number of reasons for inactivity (mean ± SD) | 1.16 ± 0.68 | 1.05 ± 1.04 | 1.54 ± 1.00 | 1.22 ± 0.88 | 1.21 ± 0.87 | 1.17 ± 1.05 | 0.292[B] | 0.399[B] |
| Reasons for inactivity, *n* (%) | | | | | | | | |
| No time | 43 (70.5%) | 34 (55.7%) | 30 (81.1%) | 26 (70.3%) | 53 (70.7%) | 44 (58.7%) | 0.445[A] | 0.342[A] |
| Too tired | 25 (41.0%) | 17 (27.9%) | 14 (37.8%) | 8 (21.6%) | 21 (28.0%) | 16 (21.3%) | 0.258[A] | 0.635[A] |
| No wish | 1 (1.6%) | 8 (13.1%) | 7 (18.9%) | 8 (21.6%) | 11 (14.7%) | 18 (24.0%) | 0.012[A] | 0.267[A] |
| Do not like | 1 (1.6%) | 2 (3.3%) | 2 (5.4%) | 2 (5.4%) | 3 (4.0%) | 6 (8.0%) | 0.658[C] | 0.491[C] |
| Other | 1 (1.6%) | 3 (4.9%) | 4 (10.8%) | 1 (2.7%) | 3 (4.0%) | 4 (5.3%) | 0.100[C] | 1.000[C] |
| Total number of preferred activities (mean ± SD) | 2.39 ± 1.26 | 2.68 ± 1.48 | 2.57 ± 1.25 | 2.97 ± 1.23 | 1.92 ± 1.1 | 2.79 ± 1.49 | 0.010[B] | 0.371[B] |
| Type of preferred physical activity, *n* (%) | | | | | | | | |
| Walk | 34 (55.7%) | 37 (60.7%) | 24 (64.9%) | 32 (86.5%) | 35 (46.7%) | 54 (72.0%) | 0.182[A] | 0.023[A] |
| Jogging | 18 (29.5%) | 25 (41.0%) | 12 (32.4%) | 16 (43.2%) | 18 (24.0%) | 28 (37.3%) | 0.599[A] | 0.815[A] |
| Individual gym | 15 (24.6%) | 25 (41.0%) | 8 (21.6%) | 13 (35.1%) | 12 (16.0%) | 26 (34.7%) | 0.451[A] | 0.724[A] |
| Group gym | 12 (19.7%) | 14 (23.00%) | 8 (21.6%) | 3 (8.1%) | 8 (10.7%) | 18 (24.0%) | 0.219[A] | 0.116[A] |
| Group sports | 17 (27.9%) | 18 (29.5%) | 7 (18.9%) | 14 (37.8%) | 18 (24.0%) | 25 (33.3%) | 0.604[A] | 0.693[A] |
| Swimming | 18 (29.5%) | 21 (34.4%) | 17 (45.9%) | 19 (51.4%) | 20 (26.7%) | 23 (30.7%) | 0.107[A] | 0.094[A] |
| Skiing | 13 (21.3%) | 8 (13.1%) | 8 (21.6$) | 4 (10.8%) | 16 (21.3%) | 11 (14.7%) | 0.999[A] | 0.851[A] |
| Dance | 17 (27.9%) | 11 (18.0%) | 10 (27.0%) | 9 (24.3%) | 14 (18.7%) | 14 (18.7%) | 0.394[A] | 0.719[A] |
| Other | 2 (3.3%) | 5 (8.2%) | 1 (2.7%) | 0 (0.0%) | 3 (4.0%) | 10 (13.3%) | 1.000[C] | 0.061[A] |
| Total number of planned activities (mean ± SD) | 1.61 ± 0.93 | 1.89 ± 1.05 | 1.89 ± 0.85 | 2.32 ± 1.31 | 1.51 ± 0.86 | 2.04 ± 0.97 | 0.079[B] | 0.244[B] |
| Wish for physical activity, *n* (%) | 58 (95.1%) | 53 (86.9%) | 35 (94.6%) | 36 (97.3%) | 69 (92.0%) | 67 (89.3%) | 0.855[C] | 0.232[A] |
| Satisfaction with physical activity, *n* (%) | 9 (14.8%) | 25 (41.0%) | 4 (10.8%) | 10 (27.0%) | 13 (17.3%) | 23 (30.7%) | 0.660[A] | 0.287[A] |

**Note:**
Data are presented as *n* (%), means ± standard deviation or median (IQR); BL: baseline, 12M: 12 months after intervention; p BL: *p* value between Groups 1, 2 and 3 at baseline; p 12M: *p* value between Groups 1, 2 and 3 after 12 months; * *p* < 0.001 vs. starting BL values; Group 1: Web-based intervention only; Group 2: Combined intervention with motivational interview and Web-based intervention; Group 3: without intervention.

most of the participants, without significant difference, both at baseline and 12 months after the intervention. However, the biggest percentage of those who wished to be physically active, after 12 months, is found in Group 2.

## Predictors of students' physical activities—regression models

Direct logistic regression (Table 4) was conducted to evaluate the impact of multiple factors on the probability of increasing physical activity, measured in an average change of MET. The change of MET was defined as the positive difference between the values for baseline (BL) and after 12 months (12M). Due to a large number of factors that could affect the dependent variable (change of MET), they were grouped in three models: basic socio-demographic and anthropometric data, intervention and willingness for change, and health status with life choices.

**Table 4 Logistic regression identifying associations between physical activity with type of intervention and other physical activity related variables.**

| Variables | B | S.E. | Wald | df | Sig. | Exp(B) | 95% CI for Exp(B) | |
|---|---|---|---|---|---|---|---|---|
| | | | | | | | Lower | Upper |
| Type of intervention | | | 19.581 | 2 | 0.000 | | | |
| Motivational interview and WB intervention | 1.171 | 0.694 | 2.848 | 1 | 0.091 | 3.225 | 0.828 | 12.561 |
| Without intervention | −1.344 | 0.410 | 10.741 | 1 | 0.001 | 0.261 | 0.117 | 0.583 |
| Total number of reasons for inactivity BL | 0.704 | 0.318 | 4.894 | 1 | 0.027 | 2.022 | 1.084 | 3.772 |
| Satisfaction with physical activity BL (coded 1—No) | −0.428 | 0.627 | 0.467 | 1 | 0.494 | 0.652 | 0.191 | 2.226 |
| Wish for physical activity BL (coded 1—yes) | 0.928 | 0.746 | 1.547 | 1 | 0.214 | 2.528 | 0.586 | 10.906 |
| Total number of preferred activities BL | −0.198 | 0.183 | 1.163 | 1 | 0.281 | 0.821 | 0.573 | 1.175 |
| Total number of planned activities BL | −0.140 | 0.247 | 0.324 | 1 | 0.569 | 0.869 | 0.536 | 1.409 |
| Constant | 0.619 | 0.833 | 0.552 | 1 | 0.458 | 1.856 | | |

Note:
Variable(s) entered on step 1: Type of intervention, Total number of reasons BL, Satisfaction with physical activity BL, Wish for physical activity BL, Total number of preferred activities BL, Total number of planned activities BL; BL, baseline, WB, Web-based intervention.

The only significant model was with intervention and willingness for change as factors that could affect the change of MET (Table 4). The second model contained, as independent variables, parameters related to physical activity intervention: belonging to one of three groups (Web-based intervention only, motivational interview and Web-based intervention, and without intervention), the total number of reasons for inactivity, the total number of preferred activities, planned activities, wish for physical activity and satisfaction with physical activity. The whole model is statistically significant (hi2 = 32.7, df = 7, $n$ = 170, $p$ < 0.001). The model explains between 17.5% Cox and Snells $R$-squared and 24.5% Nagelkerkes $R$-squared and accurately classifies 73.5% of cases. The sensitivity of the model is 86.2%, and the specificity is 46.3%. The positive predictive value is 77.5%.

There is a highly significant overall effect for type of intervention (Wald = 19.5, df = 2, $p$ < 0.001) with high odds for the increase of physical activity. Students in Group 2, "Motivational interview and Web-based intervention," are 3.25 times more likely than those from Group 1 "Only Web-based intervention" (reference category) to increase physical activity. This trend was not statistically significant, but students "without intervention" are 0.26 times less likely to increase physical activity, significantly different before and after controlling for other examined parameters related to physical activity. The students with a higher number of reasons for inactivity at baseline are two times more likely to improve physical activity as well.

Mixed ANOVA model for measurement times (baseline and after 12 months) found that there was a significant main effect for the three groups ($F$ = 16.41, $p$ < 0.001) on the dependent variable MET, with a measure of the relatedness of 18.2% (partial $\eta^2$ (Eta-squared) = 0,182). There was a significant general difference in MET values between the groups.

By comparing the pairs of groups via the Turkay HSD test, with the Bonferoni correction, it was found that there was a significant difference in MET change between the groups "Only WB intervention" (mean ($M$) = 7.60 and "without the intervention"

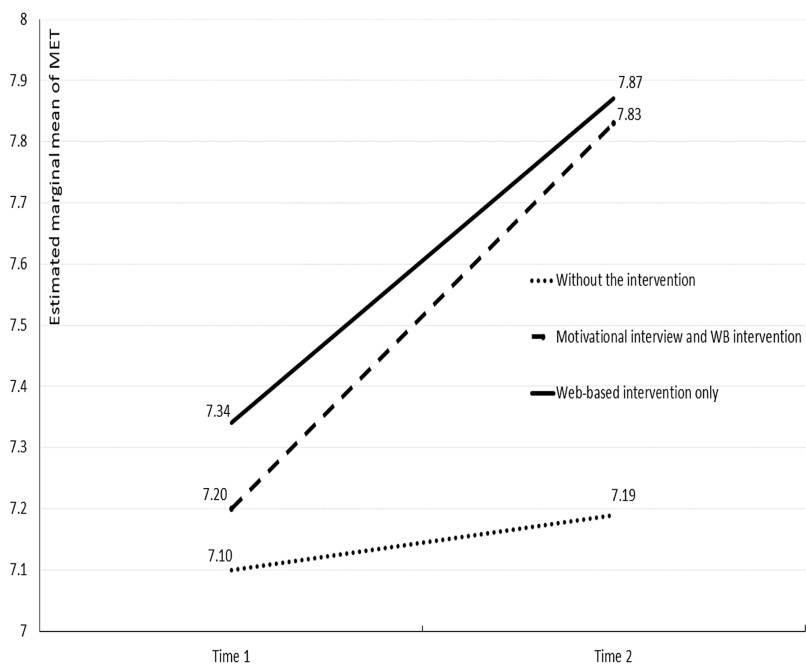

**Figure 2 Estimated marginal means of MET before and after the intervention.** MET, metabolic equivalent of task.

$M = 7.15$, $p$ <0.001) (mean difference = 0.458, 95% CI [0.252–0.663]) and the "motivational interview and WB" group ($M = 7.52$) and "without the intervention" ($M = 7.15$), $p < 0.001$ (mean difference = 0.371, 95% CI [0.145–0.597]).

Also, the independent variable "time" presented a significant main effect ($F = 42.73$, $p < 0.001$, par. $\eta^2 = 0.225$). MET values were significantly lower before ($M = 7.214$) than after 12 months ($M = 7.629$) of intervention.

Significant relationship between time and type of intervention also existed ($F = 7.33$, $p < 0.001$, partial $\eta^2 = 0.091$). Figure 2 presents the estimated marginal means of MET before and after the intervention. A comparison of the mean values of MET by groups at two times shows the increase of MET in the first two groups but not in the group "without the intervention." The influence of both factors (time and interventions) led to a change (increase) in the dependent variable MET. The significance of the interaction of these two factors shows that one variable depends on the level of the other.

## DISCUSSION

The present study examined the success of physical activity promotion among students of medicine through motivational interview and Web-based intervention, which lasted for 6 months. Our study had three groups: Group 1 exposed to Web-based intervention, Group 2 combined intervention with a Motivational Interview and Web-based intervention, and Group 3 stayed without intervention. Comparison of physical activity calculated in average MET, before and 12 months after the intervention of 6 months, has shown a significant difference between the two groups with the intervention compared

to the third group without intervention. Furthermore, our study has pointed out that motivational interview can boost the positive change in physical activity occurring among students exposed to social media intervention. We did not obtain any significant predictors of physical activity improvement assessed by direct logistic regression, except the participation in the intervention and number of reasons given for physical inactivity at baseline.

## Unmet needs of students call for improvement of physical activity and well being

Our study provides evidence of various unmet needs of university students in terms of their lifestyle, and those results could be a cause for concern for the future trend of chronic diseases among this group. Overall, only 17.3% of examined students were walking to the faculty, while the majority were using public transport. Almost 16% of the students were smoking cigarets. This finding is inline with previously published studies in other countries (*Aceijas et al., 2017*; *Thomas et al., 2019*; *Wamamili et al., 2019*).

Regarding marijuana, 5.2% of students were consumers, slightly lower than in comparable studies (*Suerken et al., 2016*; *Ayala et al., 2017*; *Candido et al., 2018*). Overall, even half of the students reported binge drinking. Those findings are consistent with some other studies regarding alcohol misuse (*Beenstock, Adams & White, 2011*), where college years were already identified as a risk period to develop substance use disorders (*Larimer, Kilmer & Lee, 2005*).

Only one-third of the examined students had healthy eating habits, measured as daily intake of fruit and vegetable. Our findings are consistent with previous studies identifying outcomes of compromised dietary balance. In a study from four European countries, only 15–32% of university students reported daily vegetable intake. In contrast, for the same group, daily fruit consumption was found in less than 50%, which is still much higher compared to our study (*El Ansari, Stock & Mikolajczyk, 2012*). Numerous studies have shown a low prevalence of fruit and vegetable intake by undergraduates (*Cooke & Papadaki, 2014*; *Farias et al., 2014*).

Regular physical activity during the time of the transition from youth to adulthood is an essential base for adult life patterns (*Telama et al., 2014*). However, at the beginning of our study, the level of physical activity was low among all groups of students, and most of them were not satisfied with such performance. This result is in line with the finding that levels of physical activity among students are not sufficient (*Marques et al., 2016*). Though physical activity improved after 12 months among students exposed to the intervention, the level of satisfaction with physical activity did not significantly change for neither of the groups.

## Reasons for the low level of physical activity and possible factors for improvement

The main reason leading to physical inactivity in our study was lack of time. This finding is in line with a similar result of a study conducted among female medical students where the most critical barrier to exercise was also lack of time (*Majeed, 2015*) and in line with a

study from 2019 (*Thomas et al., 2019*). Also, some qualitative studies with focus groups revealed that lack of time is a common if not the most common reason for physical inactivity among students (*Greaney et al., 2009*; *Nelson et al., 2009*; *Ranasinghe et al., 2016*) which was confirmed in the recently published study among the university students (*Oluyinka & Endozo, 2019*).

After 12 months after the intervention period, we found that "lack of time" remained the main reason for a low level of physical activity for all participants, even those included in the intervention.

During years at college, students are facing plenty of distractions, especially those living not at home with their families. An important skill, which needs practice, is time management. Without time management, students may find themselves behind on their studies, experiencing mental and emotional stress, or even at the risk of failing. Even though students of higher grades do report significantly better time-management skills than first- and second-year students (*Trueman & Hartley, 1996*), universities could actively try to improve basic time management strategies so that learners can improve their overall learning experience and can allocate time to physical activity.

Other factors mentioned in the literature, which can stimulate physical activity among students, are: well-being, fun and pleasure, contact with others, and health (*Diehl et al., 2018*). On the other side, university students with high sedentary behavior and students who spent more than 7 h per week studying are more likely to be physically inactive. One study has shown a significant difference in terms of gender were women had a higher chance of being physically low active (*Concha-Cisternas et al., 2018*). This result was not found in our study, as we did not obtain any significant predictors of physical activity improvement assessed by direct logistic regression, except the participation in the intervention and number of reasons for physical inactivity.

## Preferred physical activity

In terms of the type of preferred activity, there is a significant difference ($p = 0.023$) among all three groups after the intervention: walking was the preferred activity in all groups. The most significant number of students who prefer walking after 12 months was in Group 2—exposed to both Motivational Interview and Web-based intervention, where 86.5% of students appreciated the option to walk. Those students under our intervention were much more interested in walking compared toother studies (*Majeed, 2015*; *Doyle, Khan & Burton, 2019*).

The particular group, that is, medical students, could be especially interesting, as those current university students in the future probably can play an essential role as opinion leaders in establishing norms of life for the general public and their patients (*Leslie et al., 1999*; *Sehgal, 2018*).

A comparison of the mean values of MET by groups at two times shows the increase of MET in the first two groups but not in the group "without intervention." The influence of both factors (time and interventions) led to a change (increase) of dependent variable MET. The significance of the interaction of these two factors shows that one variable

depends on the level of the other. This result means that we were able to increase the frequency and intensity of physical activity in both groups under intervention.

## Strengths and weaknesses of the study

The advantage of this study is the cohort design, where both the motivational interview and web-based intervention were used to improve physical activity among university medical students. Second, the group of students of medicine is particularly interesting as, in the future, their role will be to promote a healthy lifestyle to the population. Third, the strength of our study is the examination of a wide variety of possible predictors of physical activity, which were used in the logistic models, allowing us to detect significant determinants. However, some limitations should be acknowledged.

First, the groups who accepted to participate in the study were formed based on the students' choices. The rationale behind this design is to overcome situational bias and possibly drop out as the participants' reaction to the assigned vs. voluntarily chosen intervention. We intended to introduce as many students as possible in the intervention, ensuring sufficient power to detect effects if they existed. Therefore, we have chosen approach of voluntary involvement as we did have many proofs that this group of participants is reporting lack of time as the main reason for physical inactivity, which can cause significant drop out from the intervention (*Majeed, 2015*; *Thomas et al., 2019*; *Greaney et al., 2009*; *Nelson et al., 2009*; *Ranasinghe et al., 2016*; *Oluyinka & Endozo, 2019*). At the same time, there was a need to include psychological variables such as readiness to change and self-efficacy related to physical activity. Those variables can influence the interpretation of intervention impact within the groups, and they can serve for understanding mechanisms by which intervention is likely to impact physical activity (*Shaver et al., 2019*; *Bezner et al., 2018*; *Karnes et al., 2015*). Considering all mentioned, it is important to say that for all characteristics observed, we did not find any significant difference between three groups (two with intervention and one without) at the beginning of the study. Second, the response rate 12 months after the intervention should have been higher. Third, self-assessment often can differ from the real status: in the case of physical activity, physiological indicators could confirm or disconfirm the result of the IPAQ questionnaire (*Lyden et al., 2017*). Fourth, most included predictors primarily focused on physical activity and not specifically on sedentary behaviors.

Based on the finding of the study, but also the listed limitations, we can formulate some suggestions for research in the future. First, to ensure a better response rate with a different approach to the examined group. Second, it would be recommendable to include physiological indicators obtained by real-time measurement of physical activity such as with activPAL™ instead of self-assessment done by the IPAQ questionnaire. Third, assessing stages of change, processes of change, self-efficacy, and decisional balance would be highly recommended for future studies to ensure that all aspects of the possible impact on physical activity improvement are covered. Nevertheless, our study pointed to the relevance of timely interventions for the successful promotion of physical activity among students.

## CONCLUSIONS

The study confirmed the presence of low-level physical activity among students of medicine measured by MET and possible options for its improvement based on intervention. Despite expectations, socio-demographic characteristics and life choices were not related to positive changes, but the change happened under intervention. Our study showed that interventions for improvement of physical activity among students have significant potential. Though previous studies found evidence that Web-based intervention through Facebook is beneficial for positive changes of MET, we confirmed that the involvement of combined interventions could bring better results. The desirable effects of the intervention are higher if an additional booster is involved, such as a motivational interview. Taking into account that such interventions are more expensive and less accessible for the average young population, the future steps will be to assess their cost-effectiveness.

## ACKNOWLEDGEMENTS

We would like to thank all students of the Faculty of Medicine who participated in the intervention study. We would also like to thank the staff of the Institute of Social Medicine (Faculty of Medicine, University of Belgrade) for their commitment to performing motivational interviews with students.

### Funding

This work was supported by the Ministry of Science and Technological Development of Serbia (project number 175025). The funders had no role in study design, data collection and analysis, decision to publish, or preparation of the manuscript.

### Grant Disclosures

The following grant information was disclosed by the authors:
Ministry of Science and Technological Development of Serbia: 175025.

### Competing Interests

The authors declare that they have no competing interests.

### Author Contributions

- Dubravka Mandic conceived and designed the experiments, performed the experiments, analyzed the data, prepared figures and/or tables, authored or reviewed drafts of the paper, and approved the final draft.
- Vesna Bjegovic-Mikanovic conceived and designed the experiments, performed the experiments, analyzed the data, prepared figures and/or tables, authored or reviewed drafts of the paper, and approved the final draft.
- Dejana Vukovic conceived and designed the experiments, performed the experiments, analyzed the data, prepared figures and/or tables, authored or reviewed drafts of the paper, and approved the final draft.

- Bosiljka Djikanovic conceived and designed the experiments, performed the experiments, authored or reviewed drafts of the paper, and approved the final draft.
- Zeljka Stamenkovic conceived and designed the experiments, performed the experiments, authored or reviewed drafts of the paper, and approved the final draft.
- Nebojsa M. Lalic conceived and designed the experiments, authored or reviewed drafts of the paper, and approved the final draft.

## Human Ethics

The following information was supplied relating to ethical approvals (i.e., approving body and any reference numbers):

The survey has been approved by the Ethical Board of the University of Belgrade, Faculty of Medicine (decision No: 29/IX-7, date: September 19, 2016).

## Data Availability

The raw measurements are available in the Supplemental Files.

## Supplemental Information

Supplemental information for this article can be found online at http://dx.doi.org/10.7717/peerj.9495#supplemental-information.

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
