# Peer review of "Successful promotion of physical activity among students of medicine through motivational interview and Web-based intervention"

_PeerJ, doi:10.7717/peerj.9495_

## Round 0.1 · original submission · Major Revisions

Your manuscript has been reviewed and requires modifications prior to making a decision. The comments of the reviewers are included at the bottom of this letter. Reviewers indicated that the methods section should be improved. I agree with the evaluation and I would, therefore, request for the manuscript to be revised accordingly. I would also like to suggest the following changes:

Please correct the p-values “0.000” to “<0.001”. For all tables with p-values, please use * for p<0.05 and write the phrase “*p<0.05, statistically significant” in the footnote of the table. The statistical methods section also needs revision.

Reviewer 1 ·

Basic reporting

In the abstract’s Background part, please consider strengthening your problem, gap and hook. Please put comma after “In this age”.
In the abstract’s Methods, please write “MET” in full first and then use abbreviation afterwards.
L49: Please correct the citation styling.
L64. Please consider explaining how motivational interviews are proven to increase physical activity.
L72-74: Please elaborate how social media use (e.g., Facebook) can improve physical activity.
Please consider sharing descriptive statistics for Facebook usage such as how many posts, how many comments etc. Did every student participate in the online group? What were the reactions of the students to the posts in the Facebook group.
L254: whitespace on “( p=0.005)”
L375: Please revise citation and writing.
It would be better if you could ask questions about how the time is spent when students are not active (i.e., playing games, studying, staying home, getting social).
Following literature resonates with your study, and I highly recommend citing them in your paper.
Suner A, Yilmaz Y, Pişkin B. 2019. Mobile learning in dentistry: usage habits, attitudes and perceptions of undergraduate students. PeerJ 7:e7391 https://doi.org/10.7717/peerj.7391
Mitsuhashi T. 2018. Effects of two-week e-learning on eHealth literacy: a randomized controlled trial of Japanese Internet users. PeerJ 6:e5251 https://doi.org/10.7717/peerj.5251

Experimental design

The authors used motivational study addition to web-based intervention. Did you consider supporting students with only motivational interviews? Please answer “Does motivational interviews solely can contribute to physical activity?”

The authors explained details of the intervention for Facebook. However, they did not explain the details of motivational interviews. Please consider explaining more about the motivational interviews such as how many times students attend the interviews, what was the duration per student, what is the content and motivational constructs of the interviews.

Did you measure MET scores when the interventions were completed on the 6th Month. I believe that there would be some indications about the interventions at the end of the period. Please explain how and why you set the dates for measurements.

Validity of the findings

Participants gathered from the first year medical students. The groups who accepted to participate in the study were formed based on the students’ choices. Voluntary participation may add up to bias for the result of the study. Besides, the statistical methods used in the study have assumptions on randomization and independent observations (i.e., students are classmates). Therefore, there is a violation for the assumptions of the statistical methods. Please address in your analysis section how you dealt with the violations.

Additional comments

Dear authors,
Thank you for your article entitled “Successful promotion of physical activity among students of medicine through motivational interview and Web-based intervention” submitted on PeerJ. I have reviewed your paper with interest. I would like to point out my position for you to get the most of it from my review. I have reviewed your article from the eye of a scientist in medical education expertise in e-learning. Therefore, I was not able to make extensive comments for the content on motivational interview technique. I hope my comments would be sufficient for your considerations and improvement on your study.
The authors have investigated a very important topic on physical activity among medical students. The study is very timely in our era because of internet use and other accompanying indications can hurt physical activity among young adults.
I found the study well constructed between the sections and easily read, and the use of language is also flowing in the parts.

·

Basic reporting

Some minor edits needed, see line #s referenced:
23. "Numerous evidence" - awkard, reword
26. "while" - vague
70 "this" - vague
117. Include N for Group 3?
141. "Specifically" - vague
187. "the marathon" - what marathon?
Throughout, consider using "first-year" to describe medical students rather than "first-grade".

Authors mention that readiness to change is measured, and reader needs to see this in the tables pre/post.

Notably, reviewer did not closely look at edits past the Results section because comments below may affect the interpretation of results, which would warrant changes to the discussion, etc.

Experimental design

When mentioning MotivationaI Interviewing, is it face-to-face? What number of sessions were there, and what was the duration of sessions? Clarify.


In the Facebook group, is it the case that the discussion is geared at self-reflection? What is the goal? Would help the audience to know what people in the web group are getting out of that.

Validity of the findings

Reader needs to know if data were normally distributed to know if parametric statistical analyses are appropriate. If not, non-parametric and/or corrections are needed.

To determine if either intervention group had a greater impact on PA compared to the non-intervention group - need a comparison at baseline of whether there were significant differences in group physical activity and/or readiness to change at baseline. If, for example, group 3 was not ready to change and the other two groups were - particularly given the self-selection process - it is possible that the groups were motivated to change and would have changed even without intervention.

Author needs to note that Group 3 started with lower physical activity (1595 compared to 1742 and 1773 in other groups). Is this a significant difference?

SDs for the MET data are quite large. Not sure if worth mentioning or further demonstrating that even with alot of variance the changes are significant.


Is there any expectation of a temporal impact given that med students are starting a new semester, and then workload increases as the study goes on? Presumably, this might make the physical activity less likely for everyone.

Additional comments

This paper, once revised, will contribute in a meaningful way to the existing body of literature regarding phsycial activity promotion.

It may be worthwhile to consider a few things about the intervention in this line of research:
1. In the web-based intervention, it may be worthwhile to consider whether the motivational images are expected to increase physical activity or not. There is research that shows that sometimes motivational images on social media have the opposite of the intended effect. There is a line of research about ˆfitsporation" that the authors might read, which shows that such motivational messages may cause more harm than good. Perhaps an education-only intervention with reflection would be better?

2. Consider tailoring the intervention (s) to address time-limitations of med students. That is, promote physical activity that can be built into a busy day. Most likely, this is on your radar already.

---

## Round 0.2 · Minor Revisions

I would like to thank the reviewers for their thoughtful re-reading of this manuscript. Please add a point-by-point reply to the second reviewer's comments. I agree with this evaluation and I would, therefore, request for the manuscript to be revised accordingly.

Reviewer 1 ·

Basic reporting

Revisions in the manuscript address my comments and concerns.

Experimental design

Revisions in the manuscript address my comments and concerns.

Validity of the findings

Revisions in the manuscript address my comments and concerns.

Additional comments

Dear authors,
Thank you for revising and resubmitting your work again. I have reviewed your changes and revisions. Thank you for addressing the other reviewer and my concerns and questions in your letter. Now, I believe that this article deserves publishing as is.
Yours truly.

·

Basic reporting

The authors made improvements throughout. In the abstract, consider changing "first grade" to be consistent with revisions elsewhere. In the United States, "first-year" applies to medical students, whereas "first grade" refers exclusively to roughly 6-year-old children, and it may be useful to avoid reader interpretation errors.

Experimental design

The authors made many improvements and were successful in clarifying many points, including the normalcy of the data. The methods section includes the Readiness to Change. Author comments suggest that this data/inventory are not used in the current analysis, and instead will be used later. Around this, two issues:
1) It is a bit awkward to read about an inventory and then not find results/discussions that are pertinent. If attending to this point alone, it might seem warranted to remove. However, the second issue is also related:
2) Authors have included "wish for physical activity" and "satisfaction with physical activity" to address potential confounds due to self-selection into Group 3 (as opposed to a true random assignment to the control condition). Without readiness to change data and possibly even self-efficacy data, it seems possible there are confounds beyond "wish for" or "satisfaction" with physical activity. For example, it is possible that systematically - group 3 includes people who WISH for physical activity just as much as those in group 1, and also are not as ready to change their behavior (as measured by the readiness to change) and/or do not believe they are capable of physical activity (could be measured by a physical activity self-efficacy inventory).

Validity of the findings

Given the comments provided in "experimental design", the major issue is that besides justifying the method of group assignment, an abundance of caution needs to be used with the interpretation of intervention impact based on group comparisons. The potential for confounds possibly can not be directly addressed in this study, and authors need to be transparent about this in the limitations. It is possible, for example, that the students who increased Physical activity in the intervention groups did so because of their own readiness to change and self-efficacy and would have done so without the intervention. Since motivaional interviewing targets self-efficacy, measuring self-efficacy may be useful in future studies.

Additional comments

Thank you for the opportunity to review this paper. Your contributions have potential value for first-year medical students.

---

## Round 0.3 · accepted · Accept

The authors addressed the reviewer's concerns and substantially improved the content of MS. So, based on my own assessment as an editor, no further revisions are required and the MS can be accepted in its current form.

·

Basic reporting

No comment.

Experimental design

No comment.

Validity of the findings

No comment.

Additional comments

Thank you for attending to prior comments. I believe this article now appropriately discloses limitations regarding self-efficacy so that the audience can make sound interpretations of results. I support the publication of this paper.